



# Orbital-Radar v1.0.0: A tool to transform suborbital radar observations to synthetic EarthCARE cloud radar data

Lukas Pfitzenmaier[1], Pavlos Kollias[1,2], Nils Risse[1], Imke Schirmacher[1], Bernat Puigdomenech Treserras[3], and Katia Lamer[4]

[1]Institute for Geophysics and Meteorology, University of Cologne, Cologne, Germany
[2]School of Marine and Atmospheric Sciences, Stony Brook University, Stony Brook, NY, USA
[3]Department of Atmospheric and Oceanic Sciences, McGill University, Montreal, Canada
[4]Environmental and Climate Sciences Department, Brookhaven National Laboratory, Upton, NY, USA

**Correspondence:** Lukas Pfitzenmaier (l.pfitzenmaier@uni-koeln.de)

**Abstract.** The Earth Cloud, Aerosol and Radiation Explorer (EarthCARE) satellite developed by the European Space Agency (ESA) and the Japan Aerospace Exploration Agency (JAXA) launched in May 2024 carries a novel 94-GHz Cloud Profiling Radar (CPR) with Doppler capability. This work describes the open-source instrument simulator Orbital-Radar, which transforms high-resolution radar data from field observations or forward simulations of numerical models to CPR primary measurements and uncertainties. The transformation accounts for sampling geometry and surface effects. We demonstrate Orbital-Radar's ability to provide realistic CPR views of typical cloud and precipitation scenes. These results provide valuable insights into the capabilities and challenges of the EarthCARE CPR mission and its advantages over the CloudSat CPR. Finally, Orbital-Radar allows for the evaluation of kilometer-scale numerical weather prediction models with EarthCARE CPR observations.

## 1 Introduction

Spaceborne radars offer a unique opportunity to monitor clouds and precipitation globally. For instance, the National Aeronautics and Space Administration (NASA) CloudSat Cloud Profiling Radar (CloudSat CPR; (Stephens et al., 2008, 2018)) enabled several advances in cloud and precipitation physics (Rapp et al., 2013; Stephens et al., 2018; Battaglia et al., 2020b). In 2024, the next generation CPR in space will launch onboard the Earth Cloud, Aerosol and Radiation Explorer (EarthCARE) satellite (Illingworth et al., 2015; Wehr et al., 2023). The EarthCARE CPR will be the first Doppler radar in space thus providing the first set of global Doppler velocity measurements (Kollias et al., 2022). In addition to the Doppler capability, the EarthCARE CPR will have higher sensitivity than its predecessor (-35 dBZ vs. -30 dBZ) as well as a smaller footprint (0.8 km vs. 1.4 km) and shorter along track integration (500 m vs 1.1 km).

Spaceborne radars operate from platforms that orbit the Earth at speeds that exceed 7 $km^{-1}$, and employ relatively long pulses to map the vertical structure of hydrometeors in the atmosphere. The strongest echo a spaceborne radar detects is from the Earth's surface. Instrument simulators are a well-established methodology for accounting for the effects of the observing system sampling geometry on its performance (i.e., detection limit, measurement uncertainty). For example, Lamer et al. (2020)





developed an instrument forward simulator to evaluate the impact of different spaceborne CPR configurations on our ability to detect low-level clouds and precipitation using ground-based radar data as an input dataset. Along the same lines Kollias et al.
(2022) developed a forward simulator to evaluate the quality of spaceborne Doppler velocity measurements using numerical model output as input dataset. Furthermore, instrument simulators are very useful when comparing observations from different observing systems or when objectively comparing observations and models Lamer et al. (2018).

This study describes Orbital-Radar, an open-source instrument simulator that emulates the EarthCARE and CloudSat CPR instrument capabilities by transforming sub-orbital measurements from various standardized sources and Numerical Weather
Prediction (NWP) data into CPR-like observations. Orbital-Radar does not include a forward model that converts microphysical and dynamical variables from a numerical model to radar observables like other existing radar simulators (Oue et al., 2020; Mech et al., 2020). The input is radar parameters from one coordinate system (i.e., profiling cloud radar or numerical model) and the output is CPR synthetic observations. Orbital-Radar employs a combination of functions captured in the flowchart presented in Figure 1. This study demonstrates Orbital-Radar's flexibility and its use to evaluate future applications by testing
current EarthCARE CPR configurations by contrasting its results with findings from the literature on the performance of the EarthCARE CPR (Lamer et al., 2020; Kollias et al., 2022).

The study is organized as follows: Section 2 gives an overview of the datasets that can be used as input to Orbital-Radar and discusses the input data sets and their quality control. Section 3 describes the different modules of Orbital-Radar and its limitations. In section 4, example cases are presented to demonstrate Orbital-Radar's performance in scenes with shallow
convective clouds, marine stratocumulus clouds, and Arctic clouds. The summary and outlook are provided in Section 5.

## 2   Input data

The 'orbital-radar' tool transforms ground-based, airborne, or simulated NWP radar data into synthetic satellite data from spaceborne cloud profiling radar (CPR). The quality of the forward simulated CPR data strongly depends on the quality of the input data set; therefore, rigorous quality control is crucial. Furthermore, a harmonized quality assurance allows a better
comparison of calculated synthetic CPR data from different sites. 'Orbital-radar' allows several data formats form ground-based radar networks and airborne radars. The paper hadles also forward simulated radar data from Numerical Weather Prediction model output.

    o  **Ground-based radar data:** Over the last 20 years, the suborbital capabilities for atmospheric research largely increased (Lamer et al., 2023). The U.S. Department of Energy (DoE) Atmospheric Radiation Measurement (ARM) user facility
operates several fixed and mobile observatories (Kollias et al., 2020) and the European Union (EU) Aerosol, Clouds, and Trace Gases Research Infrastructure (ACTRIS, Laj et al., 2024) research infrastructure operates over 30 fixed observatories. Furthermore, the number of ground-based observatories, e.g. supersites in Jülich, Germany, (Löhnert et al., 2015), Hyytiälä, Finland, (Hirsikko, 2014), and Barbados (Stevens et al., 2016) extended by mobile observing capabilities, e.g., the Leipzig Aerosol and Clouds Remote Observations System (LACROS, Bühl et al., 2013) from the Leipzig Institute for



Tropospheric Research, TROPOS. In addition to ground-based sites, several airborne platforms with radar instruments are currently available.

   o  **Airborne radar data:** Orbital-radar supports data sets from the airborne radars Microwave Radar/radiometer for Arctic Clouds - active (MiRAC-A) onboard *Polar 5* (Mech et al., 2022; Schirmacher et al., 2023) and the Radar Airborne System Tool for Atmosphere (RASTA) onboard *Falcon* (Bouniol et al., 2008; Delanoë et al., 2013).

If the input data are from an airborne nadir-pointing radar, then the radar signal propagates into the hydrometeor layer in the same direction as that of a spaceborne radar, and no restriction to the type of cloud and precipitation systems is necessary.

   o  **Numerical Weather Prediction data input:** This is also applicable to input radar data from a numerical model forward modelled to radar observations. In this case, a forward radar operator such as the Passive and Active Microwave radiative

TRAnsfer tool (PAMTRA; Mech et al. (2020)) and the Cloud-Resolving Model Radar Simulator (CR-SIM; Oue et al. (2020) is required to convert the model variables to radar parameters. In this case, the forward radar operator can apply the appropriate direction (top-down) two-way $94$ GHz attenuation due to hydrometeors and gases. Hereafter, surface and airborne radar and numerical model data are called suborbital data.

Orbital-Radar is capable of ingesting data from several standardized data formats of vertically-pointing radar data including

those from the ESAs Generic Earth Observation Metadata Standard (GEOMS), the ACTRIS research infrastructure project, and the DOE ARM user facility (Kollias et al., 2005, 2007). Airborne data sets from MIRAC-A and RASTA are supported.

The optimal use of the tool requires quality-controlled radar data as input (Mech et al., 2022; Schirmacher et al., 2023; Bouniol et al., 2008; Delanoë et al., 2013). If the input radar data are from a $35$ GHz radar system, then, the technique described in Protat et al. (2010) is used to convert them to $94$ GHz and the same dielectric constant ($k = 0.75$) is used to

estimate radar reflectivity (Ze). This is mainly used for the ACTRIS data sets and will be applied during the data preparation of orbital radar.

The GEOMS datasets are corrected for gas attenuation using the ACTRIS data product (Tukiainen et al., 2020) which can be selected in the code and will also be applied during the data preperation step of the tool. In contrast, the ARM ARSCL contains a radar dataset already corrected for gas attenuation. The gaseous attenuation is straightforward and requires only knowledge of

the vertical profile of water vapour that can be retrieved from an atmospheric sounding (Liebe and Layton, 1987). Knowledge of the hydrometeor phase, mass, density, and number concentration is needed for the estimation of the hydrometeors attenuation. These microphysical parameters are not available from ground-based radar observations. As a result, the surface-up and space-down view of strongly attenuating cloud and precipitation systems is very different and the comparison of these views using Orbital-Radar is not recommended. If the input data are from a ground-based radar system, they should be restricted to cases

with limited attenuation such as ice clouds and shallow systems.

If the input data are from an airborne nadir-pointing radar, then the radar signal propagates into the hydrometeor layer in the same direction as that of a spaceborne radar, and no restriction to the type of cloud and precipitation systems is necessary.





## 3 Spaceborne CPR forward simulator

The core components of Orbital-Radar have been separately described in Tanelli et al. (2002); Kollias et al. (2014b); Lamer
et al. (2020); Kollias et al. (2022). These are i) the introduction of the Earth's surface radar reflectivity, ii) the application of
the CPR antenna pattern weighting function, iii) the application of the CPR range weighting function considering the details
of the transmitter pulse characteristics and the CPR receiver characteristics, iv) the along-track integration, v) the estimation
of the Doppler velocity errors, vi) the estimation of the NUBF effects on the CPR radar reflectivity and Doppler velocity, and
vii) the estimation of the CPR signal-to-noise ratio ($SNR$), which determines the random error in the CPR radar observables
along with the along-track integration. The following sections describe the transformations and assumptions in Orbital-Radar.
In the following we describe how they are implemented and treated within the orbital-radar tool.

### 3.1 Simulation of synthetic CPR data

This section describes the processes depicted in the central dashed box in Figure 1. All technical specifications of the Earth-
CARE and Cloudsat CPR mentioned below are listed in Table 1.

- **Data preparation, coordinate conversion**: Ground-based observations are typically recorded as a function of time and
  range, i.e., height above ground. Orbital-radar converts time ($t$) to along-track distance ($d$) by assuming a horizontal
  wind speed ($v_h$):

$$d = v_h \cdot t. \tag{1}$$

  The range is converted to height above ground by simply adding the surface elevation. Using a mean horizontal wind
  for the entire depth of the atmosphere that contains the radar observations is often not a good approximation given the
  variability of the wind magnitude and direction with altitude. At the same time, the profile of the hydrometeor layers
  observed by the ground-based radar captures the actual vertical structure of hydrometeors and should be altered. In the
  case of airborne and model data, the coordinates are already along-track distance and height.

- **Data preparation, introduction of surface echo**: The magnitude and vertical extent of the Earth's surface radar echo
  determine the "effective" sensitivity of the CPR in the lowest kilometre of the atmosphere (Lamer et al., 2020). The
  normalized (per unit of area) cross-section of the Earth's surface $\sigma_0$ [m$^{-1}$] represents the magnitude of the Earth's
  surface echo. Over an ocean surface, the normalized cross section is calculated using the relationship from Li et al.
  (2005) as a function of the near-surface wind speed provided in the X-MET data product. At 94-GHz, the ocean surface
  $\sigma_0$ varies between 16 to 6 dB for near-surface wind speeds of 2 to 20 ms$^{-1}$ respectively (Tanelli et al., 2008). At 94-
  GHz, the ocean $\sigma_0$ has negligible dependency on salinity or air temperature. Here, the Li et al. (2005) parameterization
  is used to model the $\sigma_0$ as a function of near-surface for a nadir pointing CPR. Orbital radar is currently optimized for
  overland is used, therefore the calculation of optimized $\sigma_0$ as a function of wind information is not implemented for data
  sets. Instead, a fixed value is used. Overland, $\sigma_0$ exhibits large variability due to its dependency on vegetation, surface

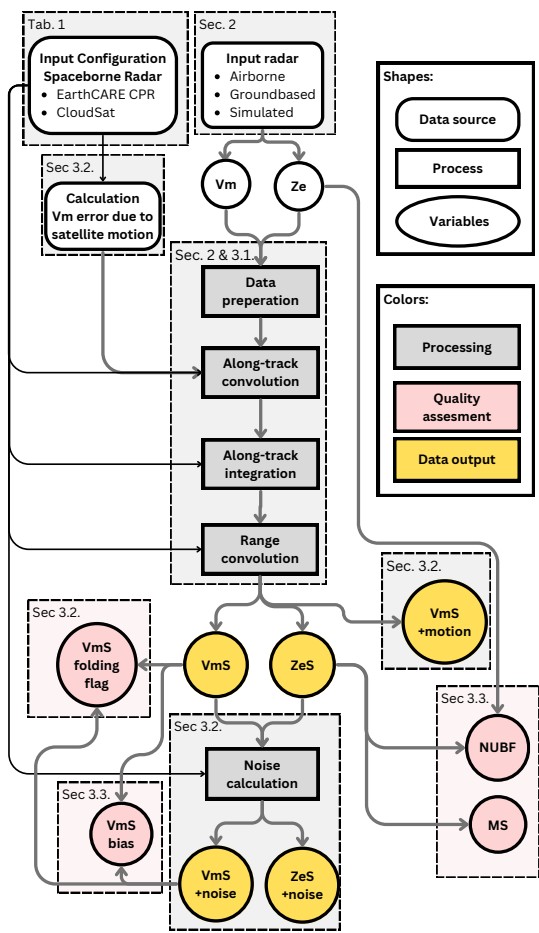

**Figure 1.** Flowchart of Orbital-Radar. Dashed boxes reference the respective Sections and Tables. The variables are radar reflectivity ($Ze$), Doppler velocity ($Vm$), noise-free CPR radar reflectivity ($Ze_S$), noise-free CPR Doppler velocity ($Vm_S$), CPR Doppler velocity with satellite motion ($Vm_S + motion$), CPR radar reflectivity with noise ($Ze_S + noise$), Doppler velocity with satellite motion and noise ($Vm_S + noise$), multiple scattering ($MS$), non-uniform beam filling ($NUBF$), folding flag ($Vm_S\ folding\ flag$), and Doppler velocity bias ($Vm_S\ bias$).

slope, soil moisture, snow cover, and other factors (Haynes et al., 2009). Thus, a fixed value of 52 dB is used. However, the user can change the value depending on the regional statistics of $\sigma_0$ or for overseas scenes. The reflectivity value of the surface echo is simulated by introducing a ground echo into the original measurements. Therefore a Gaussian distribution is added to the measurements with its peak at the range bin below the surface a peak of $\sigma_0$ and a width of $z_{res}$. The correct representation of the surface echo is performed by the along-range convolution. The range weighting function allows us to reproduce the vertical structure of the Earth's surface echo, thus creating the radar blind zone near the surface.





– **Along track convolution (spatial filtering)**: The three-dimensional pattern of the CPR pulse is described by the antenna gain weighting function $W_{ant}(x,y)$ where $x$ and $y$ represent the distance from the line of sight in the cross-radial direction and the range weighting function $W_{range}(r)$ where $r$ is the distance from the center of the CPR pulse along the radial direction (Kollias et al., 2014b; Tanelli et al., 2002). Cross-track effects are not represented in Orbital-Radar since the ground-based and airborne radar datasets are two-dimensional (time and height). Therefore, Orbital-Radar assumes cross-track homogeneity for all inputs. The $W_{(x)}$ for CloudSat is given by:

$$W_x(x) = \exp\left\{-2 \cdot \ln(2)\left(\frac{x}{0.5 \cdot IFOV}\right)^2\right\} \quad , \tag{2}$$

where $x$ is the along-track distance between suborbital observation and CPR line of sight, and $IFOV$ is the CPR instantaneous field of view (Table 1).

– **Along range convolution**: The range weighting function $W_r(r)$ depends on the transmitted waveform. The EarthCARE and CloudSat CPRs transmit a 3.3 $\mu s$ unmodulated pulse and $W_{range}(r)$ is given by:

$$W_r(r) = \exp\left\{-C_{wr} \cdot r^2\right\} \quad , \tag{3}$$

where $r$ is the distance between suborbital observation and CPR pulse centre, and $C_{wr}$ is the range weighting constant (Table 1). However, the transmitted pulse shape and frequency modulation are not the only parameters determining the detailed shape of the $W_{range}(r)$. The EarthCARE CPR uses a receiver filter that generates a sharp cut of the range side-lobes in heights above Earth's surface (Lamer et al., 2020). Therefore, the range weighting function for the EarthCARE CPR is imported from a text file. The $W_r(r)$ and $W_x(x)$ describe the instantaneous spatial filter of the CPR and are used to estimate the CPR reflectivity $Ze_{EC}$ and Doppler velocity $V_{EC}$ using the methodology described in Kollias et al. (2023a); Donovan et al. (2023).

– **Along-track integration**: In addition to the radar spatial filtering, the integration of the radar signal in the along-track introduces a temporal "stretching" filter. The integration of the convoluted data is performed according to the CPR along-track integration length ($x_{int}$; see Tab. 1).

– **Radar detection**: The minimum detectable signal ($MDS$) of the CloudSat and EarthCARE CPRs is determined by the CPR receiver noise ($N$) and the number of integrated radar samples $M$ to estimate a CPR profile. The CPR receiver noise ($N$) is reported in dBZ units to facilitate the comparison with the radar reflectivity of clouds and precipitation (Table 1). The $N$ values for CloudSat and EarthCARE are -15 and -21.5 dBZ, respectively. Using the $N$ values and the received signal $S$ strength (in dBZ), the Signal-to-Noise ratio ($SNR$) can be estimated. The $SNR$ is used in the next section to estimate the uncertainty of the CPR measurements. For a $PRF = 7000 Hz$ and 500 m along track integration, the EarthCARE CPR uses $M = 486$ samples per estimate, and for a $PRF = 4300 Hz$ and 1100 m along track integration, the CloudSat CPR uses $M = 656$ samples per estimate. The integration of $M$ samples suppresses the variance of the CPR receiver noise and allows the detection of weak signals at negative SNR values. The CloudSat $MDS$ is set to



2id="2" />

**Table 1.** Parameters for transforming suborbital radar data to synthetic CPR data (Stephens et al., 2008; Kollias et al., 2014b; Lamer et al., 2020).

| Name | Variable | EarthCARE | CloudSat |
|---|---|---|---|
| Frequency | $\nu$ | 94.05 GHz | 94.05 GHz |
| Satellite velocity | $V_{sat}$ | 7200 km s$^{-1}$ | 7000 km s$^{-1}$ |
| Satellite altitude | $h_{sat}$ | 400 km | 720 km |
| Antenna diameter | $d_{ant}$ | 2.5 m | 1.85 m |
| Pulse length | $r_{pul}$ | 500 m | 480 m |
| Vertical resolution | $z_{res}$ | 100 m | 240 m |
| Along-track integration length | $x_{int}$ | 500 m | 1100 m |
| Pulse repetition frequency | PRF | 5000 Hz | 4000 Hz |
| Noise floor | N | -21.5 dBZ | -15 dBZ |
| Minimum detectable signal | MDS | -35 dBZ | -30 dBZ |
| Surface echo equivalent reflectivity | $Ze_{z_0}$ | 52 dBZ | 52 dBZ |
| Nyquist velocity | $v_{nq}$ | $\frac{\lambda \cdot PRF}{4}$ | - |
| Antenna beam width | $\theta_{track}$ | $\frac{74.5 \cdot \lambda}{d_{ant}}$ | $\frac{67 \cdot \lambda}{d_{ant}}$ |
| Range weighting constant | $c_{wr}$ | asymmetric | $\frac{\pi^2}{2 \cdot \ln(2) \cdot r_{pul}^2}$ |
| Instantaneous field of view | $IFOV$ | $h_{sat} \cdot \tan\{\frac{\pi \cdot \theta_{track}}{180°}\}$ | |
| Wavelength | $\lambda$ | $\frac{c}{\nu}$ | |
| Speed of light | c | 299792458 m s$^{-1}$ | |

-30 dBZ and for EarthCARE is set to -35 dBZ. The $MDS$ values are valuable for estimating which parts of cloud and precipitation systems are detected by the CPRs.

## 3.2 CPR measurement uncertainty

The uncertainty in the CPR reflectivity is estimated using Hogan et al. (2005) and Delanoë and Hogan (2010).

$$\Delta Z = \frac{4.343}{\sqrt{M}}(1 + \frac{N}{S}) \tag{4}$$

where $\Delta Z$ is the standard deviation of the CPR radar reflectivity, $M$ is the number of samples and and $\frac{N}{S}$ is the noise-to-signal ratio in linear units. The CPR reflectivity errors for EarthCARE and CloudSat calculated with Equation 4 are shown in Table 2.

The $\Delta Z$ is subsequently used to add noise to the simulated CPR reflectivities:

$$Ze_{+noise} = Ze + \Delta Z_{EC} \cdot \Gamma_{Ze}, \tag{5}$$

where $\Gamma_{Ze}$ is a Gaussian distributed random number ($\mu = 0$, $\sigma = 1$, and $\Gamma_{Ze}(x)$ with $x \in [-3,3]$ (Kollias et al., 2022; Donovan et al., 2023)). In the final step, all data points below the noise floor of CPR $N$ are filtered out from the data.

2





**Table 2.** Parameters for the calculation of $Ze$ and $v_m$ noise as a function of radar reflectivity. $\Delta Z_{CS}$ is based on Hogan et al. (2005) and std($V_{TOTAL}$) is based on Kollias et al. (2022).

| $Ze$ in dB | $\Delta Z_{CS}$ in dB | $\Delta Z_{EC}$ in dB | std($V_{TOTAL}$) in $\mathrm{m\,s^{-1}}$ |
|---|---|---|---|
| -37 | - | 7.18 | 3.27 |
| -34 | - | 3.69 | 3.12 |
| -31 | 6.92 | 1.94 | 2.83 |
| -28 | 3.55 | 1.06 | 2.35 |
| -25 | 1.85 | 0.62 | 1.63 |
| -22 | 1.01 | 0.39 | 1.09 |
| -19 | 0.58 | 0.28 | 0.76 |
| -16 | 0.36 | 0.21 | 0.59 |
| -13 | 0.24 | 0.18 | 0.52 |
| -10 | 0.18 | 0.16 | 0.49 |
| -7 | 0.14 | 0.15 | 0.48 |
| $\geq$-4 | 0.1 | 0.13 | 0.47 |

The satellite velocity $V_{sat}$, antenna pointing knowledge and the presence of NUBF conditions within the radar sampling volume can lead to biases and uncertainty in the EarthCARE CPR Doppler velocity estimation (Tanelli et al., 2005; Battaglia and Kollias, 2015; Kollias et al., 2022). In Orbital-Radar, the EarthCARE CPR Doppler velocity estimation accounts for the CPR spatial volume filter and along-track integration. Furthermore, every suborbital radar point within the CPR sampling volume has been assigned an apparent Doppler velocity $V_x$ given by:

$$V_x = -x \cdot \frac{V_{sat}}{h_{sat}}, \tag{6}$$

where $V_{sat}$ is the satellite velocity, $h_{sat}$ is the satellite orbit height and $x$ is the along-track distance of the suborbital radar point from the line of sight. The introduction of $V_x$ permits the estimation of the NUBF-induced velocity bias $V_{NUBF}$ and it is reported in the output file. Using the methodology described in Kollias et al. (2023a), the $V_{NUBF}$ is removed using the along-track CPR reflectivity gradient $\Delta_x Z$. Due to uncertainty in the detail along-track CPR reflectivity structure, the NUBF correction is not perfect and an error term is introduced:

$$std(V_{NUBF}) = 0.15\ \mathrm{m\,s^{-1}} \cdot \frac{\Delta_x Z}{3\ \mathrm{dB}Z} \tag{7}$$

is a result based on statistics from EarthCARE CPR simulations using realistic numerical model scenes and actual cloud observations (Kollias et al., 2022). Equation 7 suggests that the uncertainty in the removal of the $V_{NUBF}$ velocity bias is proportional to the along-track CPR reflectivity gradient $\Delta_x Z$. In typical cloud and precipitation conditions, the median value of $\Delta_x Z$ is approximately 3 $dBkm^{-1}$ (Kollias et al., 2014a), however, in convective clouds can exceed 10 $dBkm^{-1}$. In addition to $std(V_{NUBF})$, the satellite velocity $V_{sat}$ broadens the Doppler velocity distribution (Eq. 6) within the CPR sampling volume,





thus, introduces another uncertainty term $std(V_{BROAD})$ (Battaglia et al., 2020a; Kollias et al., 2022, 2014b). The magnitude
of $std(V_{BROAD})$ depends on $M$ and $SNR$. Numerical simulations of time series of radar signals with the same characteristics
like those expected from the EarthCARE CPR have been used to estimate a $std(V_{BROAD})$ lookup table (Table 2). Finally,
these two error terms are combined to estimate the total CPR Doppler velocity uncertainty These two terms are combined to
provide the total CPR Doppler velocity uncertainty $std(V_{DOP})$:

$$std(V_{DOP}) = \sqrt{std(V_{NUBF})^2 + std(V_{BROAD})^2} \tag{8}$$

Calculation of the synthetic CPR Doppler velocity uncertainty ($V_{+noise}$) is added on top of the synthetic noise-free CPR with
satellite motion contribution data $V_{EC}$ as follows,

$$V_{+noise} = V_{EC} + std(V_{DOP})\,\Gamma_{Vm} \tag{9}$$

where $\Gamma_{Vm}$ is a Gaussian distribution of random number ($\mu = 0$, $\sigma = 1$, and $\Gamma_{Vm}(x)$ with $x \in [-v_{nq}, v_{nq}]$) representing the
general Doppler velocity error statistic of the satellite.

### 3.3 Quality flags of synthetic CPR data

Orbital-radar produces several diagnostic parameters and flags to help the user assess the quality of the simulated CPR data.
The input suborbital data have higher resolution than the CPR simulated data, thus, with the use of the CPR spatial filters we
can provide estimates of the NUBF conditions within the CPR sampling volume. The NUBF effects are amplified in areas with
significant changes in the microphysics and dynamics and near cloud edges (Pfitzenmaier et al., 2019). The standard deviation
of the radar reflectivity field within the CPR sampling filter $std(Ze)$ is used to characterize the representiviness of the simulated
CPR radar reflectivity see Section 4.1, Figure 2 f). In addition, the NUBF Doppler velocity bias $V_{NUBF}$ and the $std(V_{NUBF})$
measure of the impact of NUBF in the CPR Doppler velocity estimate.

The MS flag calculation is based on the method from Battaglia et al. (2008). The flag highlights all bins in which MS plays
a role. The calculation uses an MS scattering threshold of $12\,\mathrm{dB}$. The flag highlights the profiles affected by MS and provides
help for the interpretation of the data.

Finally, the Doppler velocity folding flag identifies CPR data where the simulated Doppler velocity exceeds the Nyquist
velocity of the EarthCARE CPR.

## 4  Application

This section demonstrates the application of Orbital-Radar to four observed or simulated suborbital cloud and precipitation
scenes. The first two scenes cover ground-based observations of shallow convective clouds (Sect. 4.1) and marine stratocumu-
lus clouds (Sect. 4.2). The latter two scenes cover airborne observations (Sect. 4.3) and numerical model simulations (Sect. 4.4)
of Arctic mixed-phase clouds. Orbital-Radar transforms the suborbital data to EarthCARE CPR observations with the specifi-
cations in Table 1. We use a mean horizontal wind of $6\,\mathrm{m\,s^{-1}}$ for ground-based and numerical model input.





### 4.1 Ground-based: Shallow convective clouds

The first case study presents shallow convective clouds observed by the ground-based 94 GHz radar MiRAC-A at the Jülich Observatory for Cloud Evolution (JOYCE; Löhnert et al., 2015) in Jülich, Germany, on 6 April 2021 (Fig. 2). Snow and graupel were detected and near-surface air temperatures were about 0°C on this day. MiRAC-A observed radar reflectivities above 15 dBZ inside the convective cores and Doppler velocities up to $2\,\mathrm{m\,s^{-1}}$ in updraft regions. We expect attenuation of the radar signal by frozen hydrometeors only due to the absence of a melting layer and the cold near-surface air temperatures.

Figure 2c and Figure 2d illustrate the impact of the EarthCARE CPR sampling volume on the small-scale reflectivity and Doppler velocity features observed by MiRAC-A. These findings are consistent with previous studies (Burns et al., 2016; Lamer et al., 2020). Figure 2e shows the EarthCARE CPR Doppler velocity with NUBF and satellite motion effects, and velocity folding due to the narrow Nyquist velocity. The simulated CPR Doppler velocity illustrates the challenges related to the measurement of convective motions from space (Kollias et al., 2022). For example, although the updrafts detected by the high-resolution ground-based radar are visible in the convoluted and integrated mean Doppler velocity (Figure 2d, the identification of the updraft regions is far more challenging in the Doppler velocity field (Figure 2e. The three diagnostics indicate NUBF near cloud edges and convective cores (Fig. 2f), multiple scattering for the cloud at 270 km (Fig. 2g), and Doppler velocity folding for few scattered range gates near cloud edges (Fig. 2h).

### 4.2 Ground-based: Marine stratocumulus clouds

The second case study using ground-based radar observations is marine stratocumulus (Figure 3). The measurements were obtained by the 94 GHz National Institute of Research and Development for Optoelectronics (INOE) radar during ASKOS campaign in Mindelo, Cape Verde, on 15 July 2022 (Marinou et al., 2023).

Figure 3a shows the ground-based radar reflectivity of the stratocumulus clouds. The cloud layer is less than 250 m thick and drizzle appears below the cloud base early in the along-track segment. Figure 3b illustrates the vertical stretching of the cloud layer due to the 500 m pulse length of EarthCARE CPR. This pulse length also causes a surface echo up to 500 m above ground (Burns et al., 2016). Figure 3c shows the Doppler velocity from the ground-based radar and Figure 3d the corresponding simulated EarthCARE CPR Doppler velocity. The CPR Doppler velocity field is noisy due to the low $SNR$ and considerable NUBF conditions. Post-processing of the raw CPR Doppler velocities can lead to substantial improvement of their quality (Sy et al., 2014; Kollias et al., 2014b, 2023b).

### 4.3 Airborne: Arctic mixed-phase clouds

Figure 4 shows the 94 GHz measurements from MiRAC-A on board the *Polar 5* aircraft during the AFLUX campaign near Svalbard, Norway (Mech et al., 2022; Schirmacher et al., 2023). The airborne MiRAC-A does not provide Doppler velocities due to the 25 ° off-nadir view of the antenna (Mech et al., 2019). The synthetic $Ze_{EC+noise}$ captures the features of the input radar reflectivity. However, a smoothing effect occurs around the cloud edges. Additionally, the ground echo covers the precipitation near the surface. One should note that the along-track resolution of the input data set is coarser than that of ground-



**Figure 2.** Shallow convective clouds observed by the ground-based MiRAC-A radar at JOYCE in Jülich, Germany, on 6 April 2021 and transformed to EarthCARE CPR with Orbital-Radar. Panels show a zoom of the 24-hour measurements of the (a) input radar reflectivity with artificial surface echo, (b) synthetic CPR radar reflectivity, (c) synthetic CPR radar reflectivity including noise, (b) input Doppler velocity, (d) synthetic CPR Doppler velocity, (e) synthetic CPR Doppler velocity including satellite motion, noise, and folding, (f) NUBF estimate, (g) MS flag, and (h) folding flag.



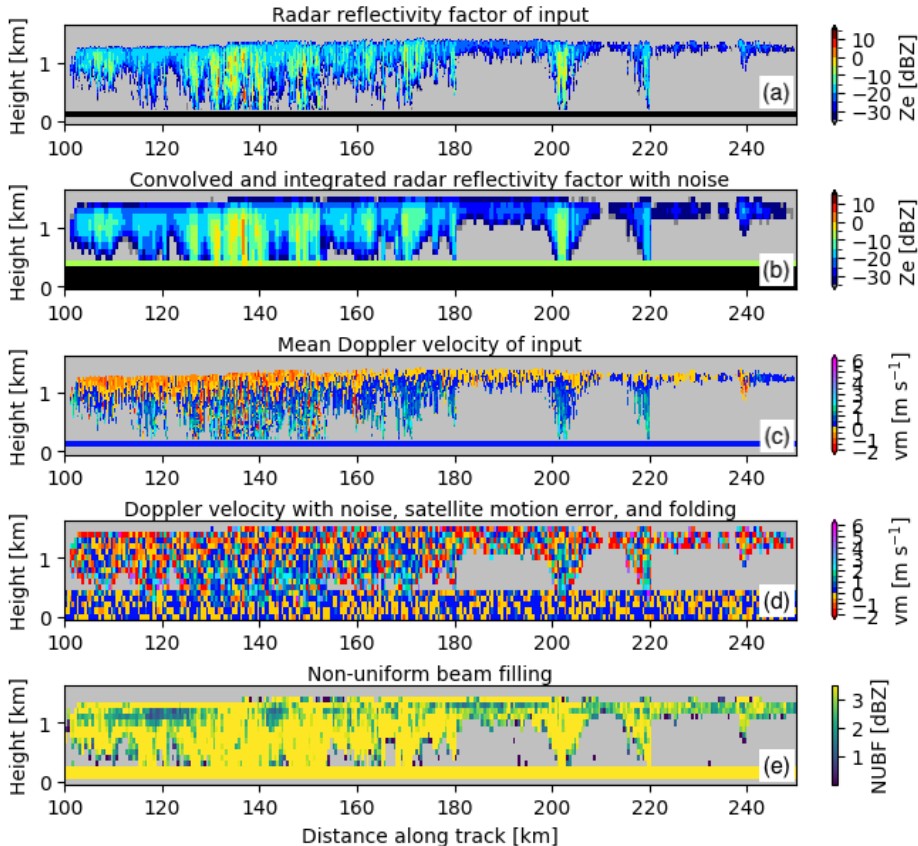

**Figure 3.** Marine stratocumulus clouds observed by the ground-based INOE radar during ASKOS in Mindelo, Cape Verde, on 15 July 2022 and transformed to EarthCARE CPR with Orbital-Radar. Panels show the (a) input radar reflectivity with artificial surface echo, (b) synthetic CPR radar reflectivity including noise, (c) input Doppler velocity, (d) synthetic CPR Doppler velocity including satellite motion, noise, and folding, and (e) NUBF estimate

based radars. This generally results in less resolved cloud structures and lower values of the NUBF estimations in the lowest 0.8 km for larger distance 80 km along-track (b) where large gradients in the $Ze$-fields are visible (a). This case demonstrates the successful transformation of airborne radar data into synthetic CPR data to study satellite overflights (Schirmacher et al., 250  2023).

### 4.4 Numerical model: Arctic mixed-phase clouds

The last case presents an Arctic cloud system simulated with a numerical model and converted to radar observation space with the forward operator PAMTRA (Mech et al., 2020). The comparison of these forward simulations with radar observations allows for an evaluation of the simulated microphysical processes in the numerical model (Ori et al., 2020). Figure 5 depicts 255  a forward-simulated scene from the high-resolution icosahedral non-hydrostatic large-eddy model (ICON-LEM; Heinze et al.,



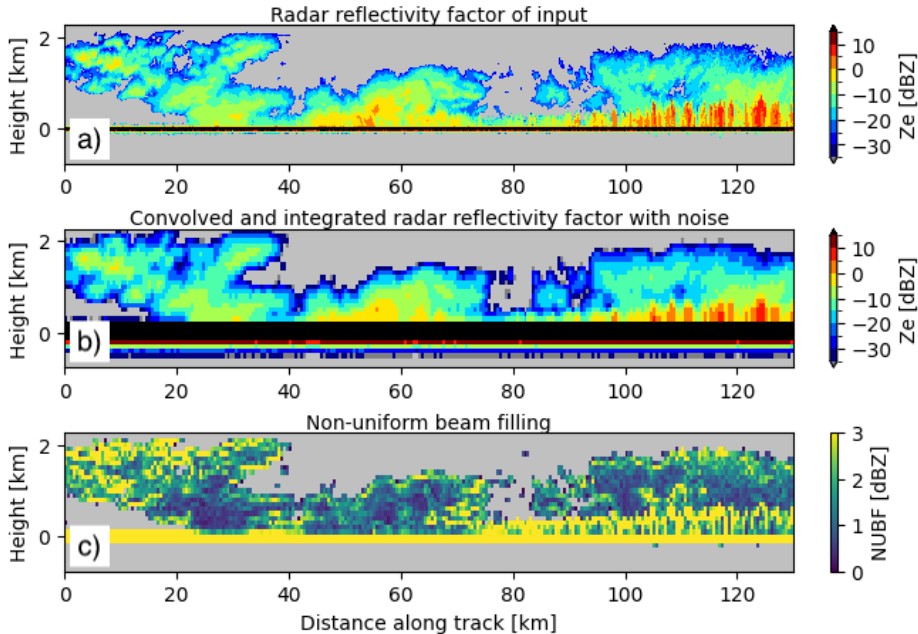

**Figure 4.** Arctic mixed-phase clouds observed by the airborne MiRAC-A radar onboard *Polar 5* during AFLUX west of Svalbard, Norway, on 1 April 2019 and transformed to EarthCARE CPR with Orbital-Radar. Panels show the (a) input radar reflectivity with artificial surface echo, (b) synthetic CPR radar reflectivity including noise, and (c) NUBF estimate

2017; Schemann and Ebell, 2020) converted to the EarthCARE CPR data using Orbital-Radar. The ICON data have coarser resolution than surface radar observations, thus, the overall comparison between the modeled and the CPR observations looks very good. Differences are only visible near cloud edges. The smooth reflectivity field also leads to a smaller NUBF contribution; only near cloud edges and regions with high radar reflectivity gradient, the NUBF effects are noticeable. Similarly,

the ICON Doppler velocity field is also smooth (Fig. 5c). After conversion to CPR, the Doppler velocity field becomes noisy due to satellite platform motion, which is the largest contributor to the CPR Doppler velocity error (Fig. 5d). In Kollias et al. (2023a), a procedure to retrieve a smooth best estimate of the hydrometeors sedimentation Doppler velocity in areas with radar reflectivity higher than -15 dBZ is described.

Using forward-modelled numerical model data as input data can have several advantages. First, model data can represent
all cloud scenarios. Using tools like PAMTRA or CR-SIM, we first need to convert the model output to radar observables at the resolution of the numerical model. In this case, the CPR hydrometeor attenuation can be directly estimated from the model output using the PAMTRA or CR-SIM forward operators. In the second step, Orbital-Radar is applied to the PAMTRA or CR-SIM output to add the sampling, sensitivity, and uncertainty effects of the spaceborne CPR. This approach can be used to evaluate the performance of future radar systems.



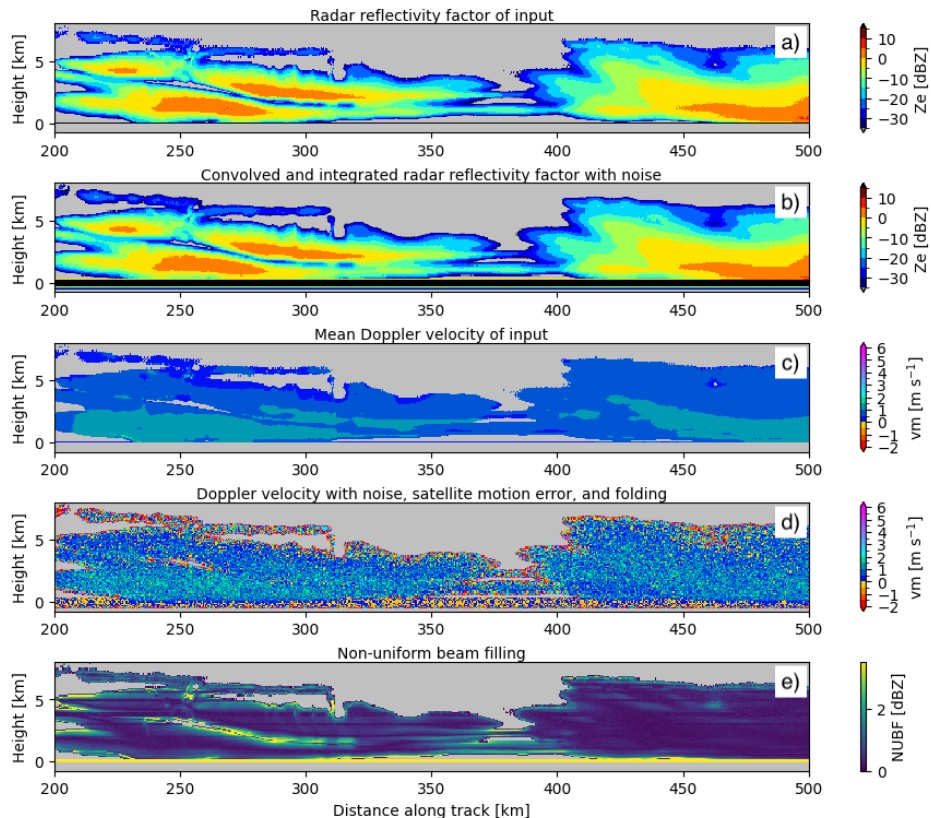

**Figure 5.** Arctic mixed-phase clouds from the NWP model ICON-LEM forward simulated to a ground-based radar with PAMTRA at AWIPEV in Ny-Ålesund, Svalbard, Norway, on 13 January 2022 and transformed to EarthCARE CPR with Orbital-Radar. Panels show the (a) input radar reflectivity with artificial surface echo, (b) synthetic CPR radar reflectivity including noise, (c) input Doppler velocity, (d) synthetic CPR Doppler velocity including satellite motion, noise, and folding, and (e) NUBF estimate.

## 5 Conclusions and outlook

This work describes Orbital-Radar, which transforms suborbital radar measurements into synthetic EarthCARE or Cloud-Sat cloud profiling radar (CPR) data. Orbital-radar used as input standardized sources of ground-based, airborne cloud radar datasets or forward-simulated radar data from numerical models. Input datasets include the European ACTRIS cloud radar network and US DOE ARM observatories. Orbital-Radar reads the different input datasets and, if needed, corrects them for gaseous attenuation and transforms the radar reflectivities from 35 to 94 GHz. In addition, if the input data are from ground-based radar, the time dimension is converted to along track distance by multiplying with a mean wind speed. The quality-controlled input datasets are used to simulate the CPRs by introducing an Earth surface echo, spatial filtering due to the antenna and range weighting functions, and along-track integration. The sensitivity of the spaceborne CPRs is also emulated with the introduction of sensor-specific noise. The introduction of noise affects the detection capability of the CPR and the un-



certainty of the key measurements. Noise is then added to the radar moments to reflect how the spaceborne CPR performance is affected by SNR, satellite motion, and NUBF. Finally, Orbital-Radar generates diagnostics to facilitate quality control, i.e., multiple scattering and Doppler velocity folding flags.

The case studies presented demonstrate that Orbital-Radar can reproduce some of the key limitations and challenges introduced by a spaceborne CPR platform. For example, the forward simulations indicate that overall the raw CPR Doppler
velocities will be noisy, and careful postprocessing is needed to enhance their quality and application for process understanding and model evaluation. Orbital-Radar facilitates direct comparisons between spaceborne radar observations and surface or airborne radar observations for validation of satellite observations. Furthermore, the tool allows for global high-resolution numerical model evaluation with spaceborne CPR observation when coupling the numerical model output with a radiative transfer model.

*Data availability.* The data used in this publication act as a test data set for the Orbital-radar tool and are available at (Zenodo, Pfitzenmaier and Risse, 2024). The data set for the ground-based w-band radar at JOYCE is downloaded from the ACTRIS database CLU (Pfitzenmaier et al., 2024), and the data set for the ground-based w-band radar at Mindelo during the ASKOS campaign is also provided by the ACTRIS database CLU (Antonescu et al., 2024). The GEOMS data filed from the ground-based w-band radars at JOYCE and Mindelo are provided by LP and stored in the database. The airborne data set from the AFLUX campaign can be found in the PANGAEA database (Mech et al.,
2022). The forward-modelled radar data using ICON output and the PAMTRA tool for the NyAlesund is provided by LP and stored in the database.

*Code and data availability.* The open-source Orbital-radar Python package is published on GitHub (https://github.com/igmk/orbital-radar, Risse, 2024). The repository also includes a Jupyter Notebook example. The data used here is available at https://uni-koeln.sciebo.de/s/amrLECxo1Ifretu.

*Author contributions.* LP and PK wrote the original manuscript draft. LP performed the data analysis. NR wrote the Python code based on previous work from LP, IS and KL. PK and BPT helped in the code design and its structure. BPT and KL participated in the review and editing of the manuscript.

*Competing interests.* The authors declare that they have no conflict of interest.

*Acknowledgements.* This work was funded, and the code was developed in the scope of ESA-funded projects. The FRM4Radar project
(Cloud profiling for Cloud Validation, Contract No. 4000122916/17/I-EF) and ACPV project, Best practice protocol for validation of Aerosol,





Cloud, and Precipitation Profiles (Contract No. 4000140645/23/I-NS). Thanks for the support from ESA and the collaboration between the University of Cologne, Stony Brook University, McGill University and Brookhaven National Laboratory. Contributions by KL were supported by the U.S. Department of Energy, Atmospheric System Research (contract: DE-SC0012704).

We acknowledge ACTRIS and Finnish Meteorological Institute for providing the data set which is available for download from https://cloudnet.fmi.fi.
We acknowledge ECMWF for providing IFS model data.



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
