# Peer review of "Orbital-Radar v1.0.0: A tool to transform suborbital radar observations to synthetic EarthCARE cloud radar data"

_Geoscientific Model Development, 2024_

## Referee Comment (RC3)

This article by Lukas Pfitzenmaier et al. presents the development and demonstration of an open-access simulator tailored for the CPR measurements of EarthCARE. The paper provides a novel way of comparing model results with observational satellite data, hence falls within the objectives of the GMD. The work is complete, scientifically accurate, and significant, and the manuscript is well-written and well-structured. Overall the study is suitable for publication. Certain sections could benefit from some additional clarifications, described herein.

L. 5-6: "*We demonstrate Orbital-Radar's ability to provide realistic CPR views of typical cloud and precipitation scenes*." It would benefit the reader If you include in the abstract some additional information of the demonstration of the realistic CPR views provided in this work. Maybe through mentioning the applications presented in the paper? Also, you could consider including in the conclusion the need of additional evaluation of the performance of the tool with CPR measurements.

Lines 74-75: "If the input radar data are from a 35 GHz radar system, then, the technique described in Protat et al. (2010) is used to convert them to 94 GHz and the same dielectric constant (k = 0.75) is used to estimate radar reflectivity (Ze)." The dielectric constant $|K^2|$ may vary in different conditions of temperature/ or different wavelengths of electromagnetic radiation (see Table 1 from Lhermitte 1989). Do you expect that using the same dielectric constant may have an impact on the results of the simulator? Also, please elaborate on the k=0.75, given that the dielectric factor used in radar meteorology is usually denoted as $K^2$.

L. 82: "*The gaseous attenuation is straightforward and requires only knowledge of the vertical profile of water vapour that can be retrieved from an atmospheric sounding (Liebe and Layton, 1987). Knowledge of the hydrometeor phase, mass, density, and number concentration is needed for the estimation of the hydrometeors attenuation. These microphysical parameters are not available from ground-based radar observations*". Can you provide additional comment on indicative cases with strongly attenuating conditions of clouds or rain where the tool phases limitations to simulate the CPR data?

L. 89: "*The core components of Orbital-Radar have been separately described.*" Can you clarify in the text if all these core components are used in the simulator?

Equation 1: Do you assume a uniform linear motion? I would suggest to elaborate a little on that.

L. 93: Please indicate that NUBF is the non-uniform beam filling. It is mentioned only in Fig 1 caption.

L. 131: typo W(x).

Lines 126-144: The authors could consider including a brief description of the concept of a weighting function, (e.g. a mathematical example like the Gaussian form $e^{-2x^2/(2\sigma^2)}$). This would help readers not very familiar with technical details, understand how the antenna gain and range weighting functions behave.

---

## Community Comment (CC2)

This article by Lukas Pfitzenmaier et al. presents the development and demonstration of an open-access simulator tailored for the CPR measurements of EarthCARE. The paper provides a novel way of comparing model results with observational satellite data, hence falls within the objectives of the GMD. The work is complete, scientifically accurate, and significant, and the manuscript is well-written and well-structured. Overall the study is suitable for publication. Certain sections could benefit from some additional clarifications, described herein.

**Thanks, Eleni Marinou, for reviewing the article and the suggestions and comments given.**

L. 5-6: "We demonstrate Orbital-Radar's ability to provide realistic CPR views of typical cloud and precipitation scenes." It would benefit the reader If you include in the abstract some additional information of the demonstration of the realistic CPR views provided in this work. Maybe through mentioning the applications presented in the paper? Also, you could consider including in the conclusion the need of additional evaluation of the performance of the tool with CPR measurements.

Thanks for the comments. The Abstract is edited and some more information included. See the edited text below or in the updated manuscript.

"... The presented case studies show small-scale convection, marine stratus clouds and arctic mixed-phase cloud cases. These results provide valuable insights into the capabilities and challenges of the EarthCARE CPR mission and its advantages over the CloudSat CPR. Finally,

Orbital-Radar allows for evaluating kilometre-scale numerical weather prediction models with EarthCARE CPR observations. So, orbital-radar can generate Cal/Val data sets already pre-launched. Nevertheless, an evaluation of synthetic CPR output data to accurate EarthCARE CPR data is missing. ..."

Lines 74-75: "If the input radar data are from a 35 GHz radar system, then, the technique described in Protat et al. (2010) is used to convert them to 94 GHz and the same dielectric constant (k = 0.75) is used to estimate radar reflectivity (Ze)." The dielectric constant |K2| may vary in different conditions of temperature/ or different wavelengths of electromagnetic radiation (see Table 1 from Lhermitte 1989).

Do you expect that using the same dielectric constant may have an impact on the results of the simulator? Also, please elaborate on the k=0.75, given that the dielectric factor used in radar meteorology is usually denoted as K2.

The code uses the transformation of the dielectric constant to calculate all data based on it. Both constants are used for w-band systems, and the values usually do not change during the radar system's operation. So, the temperature or aggregate dependency of the hydrometeors and their surrounding gases is not used to calculate their equivalent reflectivity values.

**See changes mad in the manuscript:**

"... The assumption of the transformation relies on an assumption about the mass–diameter relationship of ice particles used in the Mie scattering computations. the disparity in radar reflectivity between 35 GHz and 94 GHz begins to exceed 1 dB when the 35 GHz reflectivity reaches approximately 0 dBZ. In most cases the 35 GHz radar ice reflectivities fall below 0

dBZ. Therefore, any uncertainty arising from this approximation is deemed insignificant (Protat et al., 2010; Kollias et al., 2019). Also the the same dielectric constant (|k|2 = 0.75) is used to estimate radar reflectivity (Ze). This step is done to match the the Satellite configuration. This is mainly used for the ACTRIS data sets and will be applied during the data preparation of orbital radar. ..."

L. 82: "The gaseous attenuation is straightforward and requires only knowledge of the vertical profile of water vapour that can be retrieved from an atmospheric sounding (Liebe and Layton, 1987). Knowledge of the hydrometeor phase, mass, density, and number concentration is needed for the estimation of the hydrometeors attenuation. These microphysical parameters are not available from ground-based radar observations". Can you provide additional comment on indicative cases with strongly attenuating conditions of clouds or rain where the tool phases limitations to simulate the CPR data?

The other reviewers commented that these parts needed more information. Generally speaking, comparing ground-based and satellite CPR measurements is challenging in the presence of liquid water in the ground-based measurements. Therefore, these data are statistically compared using ice cloud data only (Protat et al., 2010; Kollias et al., 2019). This is because the attenuation correction for liquid requires additional data and information that might only be present for some of the sites and data sets. Hence, such preprocessing of the synthetic CPR data strongly depends on the additional data sets available, and no general flagging was calculated. This, on the one hand, makes the use of the synthetic data less intuitive; on the other hand, it increases the number of possible input data sets with which the tool can be used!

L. 89: "The core components of Orbital-Radar have been separately described." Can you clarify in the text if all these core components are used in the simulator?

For this comment, we edited the text. So please see the manuscript.

**"... 3. Spaceborne CPR forward simulator**

The core components of Orbital-Radar have been separately described in Tanelli et al. (2002); Kollias et al. (2014b); Lamer et al. (2020); Kollias et al. (2022) and used in the code. These are i) the introduction of the Earth's surface radar reflectivity and the response of point target into the range gates above and below the surface (effect of the oversampling of the CPR), ii) the application of the CPR antenna pattern weighting function, iii) the application of the CPR range weighting function considering the details of the transmitter pulse characteristics and the CPR receiver characteristics, iv) the along-track integration, v) the estimation of the Doppler velocity errors, vi) the estimation of the non-uniform beam filling (N U BF ) effect on the CPR radar reflectivity and Doppler velocity, and vii) the estimation of the CPR signal-tonoise ratio (SN R), which determines the random error in the CPR radar observables along with the along-track integration. The following sections describe the transformations and assumptions in Orbital-Radar. Following the flowchart (figure 1) we describe how they are implemented and treated within the orbital-radar tool. ..." Equation 1: Do you assume a uniform linear motion? I would suggest to elaborate a little on that.

Yes, a mean wind speed throughout the whole atmosphere is assumed. We know it is a really easy way to transform time into distance along a track. Nevertheless, the parametrization is an easy one; the results are robust and give stable results with a minimum of data input. See changes below or in the manuscript.

"... - Data preparation, coordinate conversion: Ground-based observations are typically recorded as a function of time and range, i.e., height above ground. Orbital-radar converts time (t) to along-track distance (d) by assuming a constant horizontal wind speed (vh) throughout the whole atmosphere: ..."

L. 93: Please indicate that NUBF is the non-uniform beam filling. It is mentioned only in Fig 1 caption.

For this comment, we edited the text. L. 131: typo W(x).

The typo was changed - please see the manuscript.

Lines 126-144: The authors could consider including a brief description of the concept of a weighting function, (e.g. a mathematical example like the Gaussian form  $e^{-2x2/(2\sigma^2)}$ ). This would help readers not very familiar with technical details, understand how the antenna gain and range weighting functions behave.

Thanks for the comment. We hope that the editing and the additional citation now available in the manuscript help readers understand the concept of weighting functions.

– Along track convolution (spatial filtering): The three-dimensional pattern of the CPR pulse is described by the antenna gain weighting function Want(x, y) where x and y represent the distance from the line of sight in the cross-radial direction and the range weighting function Wrange(r) where r is the distance from the center of the CPR pulse along the radial direction (Kollias et al., 2014b; Tanelli et al., 2002) (Donovan et al. (2023) provide an overview of the antenna pattern and the along track weighting functions to represent them in simulation). Cross-track effects are not represented in Orbital-Radar since the ground-based and airborne radar datasets are two-dimensional (time and height). Therefore, Orbital-Radar assumes cross-track homogeneity for all inputs. The Wx(x) for CloudSat is given by:

 $Wx(x) = exp\{-2 \cdot ln(2)(x0.5 \cdot IF OV)2\}, (2)$

where x is the along-track distance between suborbital observation and CPR line of sight, and IF OV is the CPR instantaneous field of view (Table 1).

– Along range convolution: The range weighting function Wr (r) depends on the transmitted waveform. The EarthCARE and CloudSat CPRs transmit a 3.3  $\mu$ s unmodulated pulse and Wrange(r) is given by:

 $Wr(r) = exp\{-Cwr \cdot r2\}, (3)$

where r is the distance between suborbital observation and CPR pulse centre, and Cwr is the range weighting constant (Table 1). However, the transmitted pulse shape and frequency

modulation are not the only parameters determining the detailed shape of the Wrange(r). The EarthCARE CPR uses a receiver filter that generates a sharp cut of the range sidelobes in heights above Earth's surface (Lamer et al., 2020). Therefore, the range weighting function for the EarthCARE CPR is imported from a text file. **Lamer et al. (2020) contains a detailed description of the effect of the range weighting function and provided us the range weighting function used in the tool.** The Wr (r) and Wx(x) describe the instantaneous spatial filter of the CPR and are used to estimate the CPR reflectivity ZeEC and Doppler velocity VEC using the methodology described in Kollias et al. (2023a); Donovan et al. (2023)

---

## Author Comment (AC3)

Reviewer 1:

This is relatively simple paper that describes a software tool to emulate EarthCARE or CloudSat measurements given ground-based, airborne, and simulated radar reflectivity and Doppler. I only have a few minor comments below to address before publication.

With this we thank the reviewer for his/her work and the suggestions made to improve our manuscript. In the following, we answer the comments and explain the changes. Please note that some answers given refer to changes made in the manuscript. If so, the explanation given in the text is short and emphasis is given on the improvements made in the text

Line 73: '*If the input radar data are from a 35 GHz radar system, then, the technique described in Protat et al. (2010) is used to convert them to 94 GHz*'. This is important. Please describe the method at a high level at least.

A longer description is added in the text, see below or in the updated Manuscript.

"…The assumption of the transformation relies on an assumption about the mass–diameter relationship of ice particles used in the Mie scattering computations. The disparity in radar reflectivity between 35 GHz and 94 GHz begins to exceed 1 dB when the 35 GHz reflectivity reaches approximately 0 dBZ. In most cases, the 35 GHz radar ice reflectivities fall below 0 dBZ. Therefore, any uncertainty arising from this approximation is deemed insignificant (Protat et al., 2010; Kollias et al., 2019). Also, the same dielectric constant ($|k|^2 = 0.75$) is used to estimate radar reflectivity (Ze). This step is done to match the satellite configuration. This is mainly used for the ACTRIS data sets and will be applied during the data preparation of orbital radar. "

Line 85: 'As a result, the surface-up and spacedown view of strongly attenuating cloud and precipitation systems is very different and the comparison of these views using Orbital-Radar is not recommended.' Are these columns flagged in the output?

The code does not contain any flags related to the attenuation of input radar data or synthetic CPR. Additional data from synergistic instrumentation must be used to estimate the influence of liquid attenuation. Better information was added to the manuscript to clarify this. See the edited text below or in the updated manuscript.

"… Since the tool only has the Ze and V m fields as input and uses no additional data or retrievals a flagging of cases with high attenuation due to liquid droplets or precipitation is not provided. Such filtering has to be done using additional information, such as Cloudnet target classification or the liquid water path (LWP) by a parallel measuring microwave radiometer. If the input data are from a ground-based radar system, they should be restricted to cases with limited attenuation, such as ice clouds and shallow systems. Nevertheless, the filtering of the data depends on the user of the data sets and might be individual and has to be specified when using the data further."

Line 119: '*Thus, a fixed value of 52 dB is used.*' Are you assuming sigma_0 = 52 dB or that the reflectivity factor is 52 dBZ? This is inconsistent wi the table.

The manuscript was updated to clarify this. See the edited text below or in the updated manuscript.

"… . Thus, a fixed value of σ0 = 52 dB is used. However, the user can change the value depending on the regional statistics of σ0 or for overseas scenes. The reflectivity value of the surface echo is simulated by introducing a ground echo into the original measurements. …"

Line 187: Bad grammar and duplicated sentence. '*Finally, these two error terms are combined to estimate the total CPR Doppler velocity uncertainty These two terms are combined to provide the total CPR Doppler velocity uncertainty std(VDOP ):*'

Thanks for this hit. The sentences were rewritten.

"… . Finally, these two terms are combined to provide the total CPR Doppler velocity uncertainty $std(V_{DOP})$. … "

Line 204: '*The MS flag calculation is based on the method from Battaglia*'. Again it's OK to cite but describe at a high level how this works.

As for the correction method above, an improved description was added to the text; see the updated manuscript.

"… . The MS flag calculation is based on the method from Battaglia et al. (2008). The MS flag using thresholds for calculating MS is present within the column. The thresholds were estimated using Monte Carlo reflectivity simulations for multiple cloud scenes and validated using CloudSat data. EarthCARE also operates at w-band, so we adopted the method, and so the flag highlights all bins in which MS plays a role. The calculation uses an MS scattering threshold of 12 dB or if the integration of the pixels from the top exceeds 42 dB. The flag highlights the profiles affected by MS and provides help for the interpretation of the data. …"

I suggest that a table is added that lists all of the variables included in the output files.

The table describing all the output data of the orbital-radar is implemented as an appendix to the paper.

" … 3.1. Simulation of synthetic CPR data

This section describes the processes depicted in the central dashed box in Figure 1. All technical specifications of the EarthCARE and Cloudsat CPR mentioned below are listed in Table 1. A table of all variables written in the netCDF output file is presented in the Appendix A, Table A1. …"

---

## Author Comment (AC4)

Reviewer 2:

Thank you, reviewer, for your comments and suggestions on our manuscript. Together with the suggestions of the other rewiewers we implemented them to the best of our knowledge into the text.

The paper presents the open-source instrument simulator Orbital-Radar. The manuscript is well written. I only have few minor comments. Comments:

Line 74: "the same dielectric constant (k = 0.75)" Do you mean  $|K|^2$ ?

**Thanks for the comment. This was a mistake now in the manuscript should be written $|k|^2$**

Line 84: "If the input data are from a ground-based radar system, they should be restricted to cases with limited attenuation such as ice clouds and shallow systems." Embedded liquid layers could cause significant attenuation of W-band radar observations. Do you recommend to use MWR or lidar observations to diagnose mixed-phase conditions? Would you recommend a LWP threshold that would define where your tool should or should not be used?

First, we edited the manuscript and provided some recommendations on how to handle attenuation. Correcting liquid attenuation in radar data isn't straightforward and usually requires additional data, such as a microwave radiometer and the retrieved liquid water path. In this code, we decided not to tackle this topic and left it up to the user to filter the data using their own post-processing, thresholds, etc.

Nevertheless, the problem of data filtering and the definition of attenuation in the ground and the CPR data are not present in the level 1 data for EarthCARE, which the orbital radar tool tries to mimic. Therefore, we should have included it in the tool. In addition, for some data sets, no additional parallel measurements or data sets are present, which would limit the possible input data set to the tool.

See the edited text below or in the updated manuscript.

"... Since the tool only has the Ze and V m fields as input and uses no additional data or retrievals a flagging of cases with high attenuation due to liquid droplets or precipitation is not provided. Such filtering has to be done using additional information, such as Cloudnet target classification or the liquid water path (LWP) by a parallel measuring microwave radiometer. If the input data are from a ground-based radar system, they should be restricted to cases with limited attenuation, such as ice clouds and shallow systems. Nevertheless, the filtering of the data depends on the user of the data sets and might be individual and has to be specified when using the data further."

Line 90 "the introduction of the Earth's surface radar reflectivity" Radar reflectivity characterizes a volume target. I am not sure how surface radar reflectivity is defined.

The surface reflectivity value we use in the simulation is based on the simulation studies and the parametrisation from (Li et al., 2005). It reflects the Ze value of the point target response

of the mean sea surface. Since EarthCARE and CloudSat are oversampling their received signals and the surface echo is usually a substantial reflecting target, the echo is affected by the weighting function, which leads to a so-called blind zone near the surface in the CPR data. This means that the surface echo present in the lowest range bins of the CPR overlays all atmospheric targets, if any are present.

Eq. 3: Do you have a reference for the EarthCARE's CPR pulse shape? Table 1. You assume PRF of 5000 Hz. What are the actual PRF values used by EarthCARE CPR?

The PFR used for the predefined EarthCARE configuration is 6000 Hz; the 5000 Hz was a typo. However, the PRF of EarthCARE varies from 6100 to 7500 Hz depending on the latitude over which the satellite is flying.

For the paper, we fixed the Nyquist velocity to 5.7 ms-1 and did not calculate the velocity range via the PRF relation. Nevertheless, the description was incorrect, and the table's value was changed.

Eq. 5. Is the reference (Kollias et al., 2022) correct? I was not able to find justification for using normally distributed reflectivity noise. As far as I remember i,q data follows normal distribution. The reflectivity factor should follow chi^2, if I am not mistaken. What are the units of Eq 5?

The noise we modelled follows a Gaußian distribution because it can be approximated as such a distribution in dB space and considering a large sample size. However, you are right; usually, it follows a Chi^2 distribution. We think the differences between the Gaußian and the Chi^2 distribution and, therefore, simply used the straightforward representation. In the future, we could also consider upgrading the representation of the noise.

The reference in dead needed to be corrected. Thanks for the hint.